# Caregivers' perception of risk for malaria, helminth infection and malaria-helminth co-infection among children living in urban and rural settings of Senegal: A qualitative study

**Muhammed O. Afolabi**[1]*, **Ndèye Mareme Sougou**[2], **Aminata Diaw**[2], **Doudou Sow**[3], **Isaac A. Manga**[2], **Ibrahima Mbaye**[4], **Brian Greenwood**[1], **Jean Louis A. Ndiaye**[4]

**1** London School of Hygiene & Tropical Medicine, London, United Kingdom, **2** Université Cheikh Anta Diop, Dakar, Senegal, **3** Université Gaston Berger de Saint-Louis, Saint-Louis, Senegal, **4** Université de Thies, Thies, Senegal

* Muhammed.Afolabi@lshtm.ac.uk

**Data Availability Statement:** All data underlying the findings described in this manuscript have

## Abstract

The parasites causing malaria, soil-transmitted helminthiasis and schistosomiasis frequently co-exist in children living in low-and middle-income countries, where existing vertical control programmes for the control of these diseases are not operating at optimal levels. This gap necessitates the development and implementation of strategic interventions to achieve effective control and eventual elimination of these co-infections. Central to the successful implementation of any intervention is its acceptance and uptake by caregivers whose perception about the risk for malaria-helminth co-infection has been little documented. Therefore, we conducted a qualitative study to understand the caregivers' perspectives about the risk as well as the behavioural and social risk factors promoting malaria-helminth co-infection among pre-school and school-age children living in endemic rural and urban communities in Senegal. In June and December 2021, we conducted individual and group interviews, and participant observations, among 100 primary caregivers of children recruited from Saraya villages in southeast Senegal and among leaders and teachers of Koranic schools in Diourbel, western Senegal. Our findings showed that a majority of the study participants in the two settings demonstrated a high level of perception of risk for malaria and acceptable awareness about handwashing practices, but had misconceptions that malaria-helminth co-infection was due to a combination of excessive consumption of sugary food and mosquito bites. Our observations revealed many factors in the house structures, toilet practices and handwashing with ashes and sands, which the caregivers did not consider as risks for malaria-helminth co-infections. These findings underscore the need to promote caregivers' awareness about the existence and risk of malaria-helminth co-infection in children. This approach would assist in addressing the caregivers' misconceptions about the occurrence of the co-infection and could enhance their uptake of the strategic interventions targeted at achieving control and subsequent elimination of malaria and helminth co-infection.

been made freely available to other researchers within this manuscript.

**Funding:** This study was implemented as part of a career development fellowship awarded to MOA, which is funded under the UK Research and Innovation Future Leaders Fellowship scheme (MR/S03286X/1). The funder had no role in the study design, data collection and analysis, decision to publish, or preparation of the manuscript.

**Competing interests:** The authors have declared that no competing interests exist.

## Introduction

Malaria remains a major health problem, especially in sub-Saharan Africa (SSA) where more than 90% of deaths due to the infection occur [1]. Complicating this high burden of malaria among children is the co-existence of intestinal and genito-urinary worms. Prominent among the worms are soil-transmitted helminths (STH), primarily hookworm (*Ancylostoma duodenale* and *Necator americanus*), roundworm (*Ascaris lumbricoides*), whipworm (*Trichuris trichiura*), *Schistosoma haematobium* and *S. mansoni* [2, 3]. Environmental and host factors play crucial roles in the overlap and co-existence of the multiple infections involving helminth and malaria parasites [4]. The additive effects of the mixed parasitic infections in children living in the poorest countries of the world lead to anaemia [5, 6], poor physical and cognitive developments [7] and preventable deaths [8].

Existing control programmes for worms are currently operating below the expected targets [9], despite the commitments and support that followed the 2012 London Declaration on Neglected Tropical Diseases (NTD), which aimed to achieve 75% treatment coverage and eliminate some NTDs by 2020 [10]. On the other hand, a malaria chemo-preventive programme, called Seasonal Malaria Chemoprevention (SMC), introduced in the same year 2012, has achieved more than 75% treatment coverage and prevented 75–85% cases of uncomplicated and severe malaria in children [11]. This encouraging development supports the need to explore the possibility of integrating worm control programmes with successful delivery platforms such as SMC. This integrated approach would align worm and malaria control with the WHO roadmap for NTD of ending the neglect to attain Sustainable Development Goals, by helping to eradicate diseases of poverty and promoting health and well-being for those at risk [9].

Given the widely advocated strategies for an integrated control of malaria and helminths, the availability of reliable data on caregivers' perception of the risk for malaria-helminth co-infection is key to successful implementation and uptake of coordinated control approaches by the caregivers of children who are at risk of malaria-helminth co-infections.

The ability of caregivers to perceive the risks for malaria and helminth co-infection is important in preventing these co-infections in their children. Similarly, awareness of the existence and harmful effects of co-infections and their clinical features are crucial factors which shape the health-seeking behaviours of caregivers [12]. Empirical studies on malaria [13, 14] have shown that delay in seeking early and appropriate care was largely due to poor perception about the risk of a child's vulnerability to the diseases by the caregivers. In other studies [15, 16], most caregivers who demonstrated acceptable health-seeking behaviour recognised the danger signs for severe forms of malaria, pneumonia, diarrhoea and meningitis. However, very few studies [15, 17] have investigated the caregivers' perception to the risk of dual or multiple infections such as malaria and helminths. Understanding the importance of malaria-helminth co-infection and its influence on health-seeking behaviour from the caregivers' perspectives is critical to the development and uptake of integrated approaches to prevention and treatment to the co-infection.

Given that the Health Belief Model (HBM) is one of the most important health behavioural theories [18], we postulated a framework for this study on a central concept in the HBM as the 'perceived susceptibility', which refers to the perceived chance of acquiring a condition. The 'perceived susceptibility' and the 'perceived severity' leads to the formation of 'perceived barrier' of a certain condition. The likelihood of performing a certain health behaviour is directly linked to the 'perceived threat', the 'perceived benefits' and 'barriers' of the suggested behaviour change, the self-efficacy and the cues to action [18, 19].

Guided by the principles of HBM, we conducted this study to understand from the caregivers' point of views as well as the behavioural and social risk factors (e.g. household sanitation

and hand-washing practices) promoting malaria-helminth co-infection among pre-school and school-age children living in the endemic rural and urban communities in Senegal. Although, our primary objective was to ascertain what the caregivers perceived about malaria-helminth co-infection, however, to ensure clarity of the primary objective, we needed first to systematically consider what the caregivers perceived to be the risks for malaria and helminths before exploring their perceptions on the co-infection involving malaria and helminths.

## Methods

### Ethics statement

This study was approved by the Research Ethics Committees of the London School of Hygiene & Tropical Medicine and Conseil national de Recherche en Santé (CNERS), Senegal. Given that the study was non-interventional and its implementation posed little or no risk to the study participants, verbal consent was considered sufficient for this study [20]. The Ethics Committees approved the verbal consent procedure for this study; and we obtained verbal consent from all potential caregivers before the interviews were conducted. The detailed information about the study was provided to a potential study participant by a trained research assistant, after the participant's concerns and questions were satisfactorily addressed, they signified consent to participate in the study orally. This process of verbal consenting was documented by the research assistant in a dedicated consent register.

We also obtained permissions from the health authorities at the national, regional and district levels in Senegal, prior to the implementation of the study. Participation in the study was voluntary and participant confidentiality was maintained.

### Study settings

We conducted this study as a qualitative component of two prospective population surveys designed to address the lack of information regarding the burden of malaria-helminth co-infections and its associated risk factors among pre-school and school-aged children in Diourbel and Saraya districts of Senegal. Diourbel in the western region of Senegal is mainly urban while Saraya in the south-eastern region of the country is mainly rural. Diourbel and Saraya are about 134 km east of and 740 km south of Dakar, the capital of Senegal, respectively (Fig 1). The two communities share similar epidemiological profiles. Diourbel and Saraya have a tropical Sudano-Sahelian climate with well-defined dry and rainy seasons that result from northeast winter winds and southwest summer winds; the dry season lasts from December to May and the rainy season from June to November. Diourbel and Saraya are parts of the communities most affected by malaria [21] and helminths [22] in Senegal. The main ethnic groups in Saraya are Malinke and Diakhanke, while Serer and Wolof are the main ethnic groups in Dioubel.

### Study population

The study population consisted of primary caregivers of children aged 1–14 years who were vulnerable to malaria-helminth co-infection in Diourbel and Saraya districts of Senegal. In Saraya site, the primary caregivers were the parents and guardians of these children. In Diourbel site, pre-school and school aged children lived as full boarders in Koranic schools (also called 'dahras') where they received Arabic and Islamic teachings. The primary caregivers of the children learning in these schools were Koranic teachers, heads of Koranic schools and their wives. Older Koranic students (Supervising Talibé) were also appointed by the heads of the Koranic schools to supervise and serve as caregivers to the younger students in the schools.

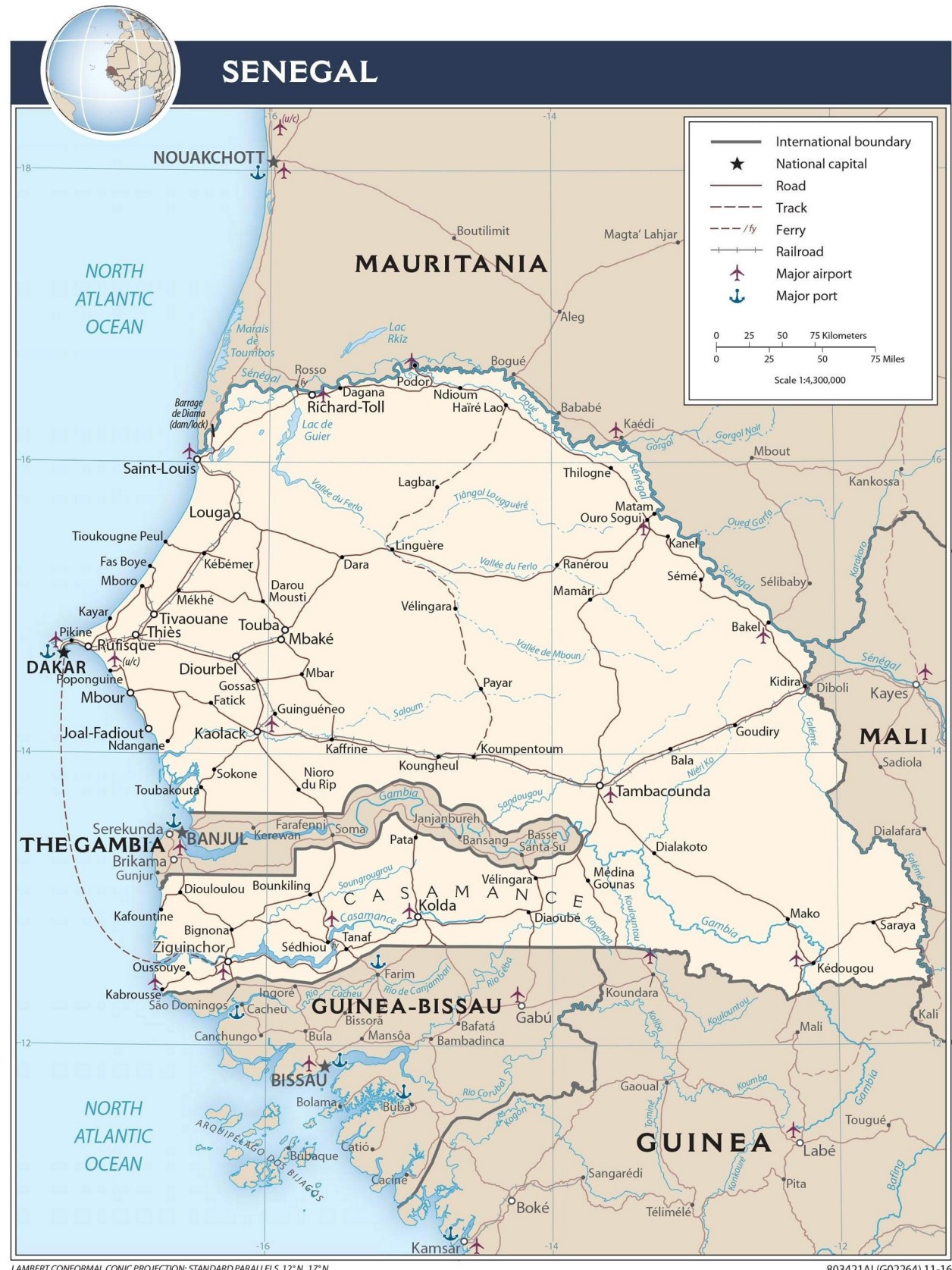

**Fig 1. Map of Senegal showing the study sites.**

Because of the differences in the settings of the two study sites, we adopted a multiple case study approach for Saraya and a single case study for Diourbel. We used a multiple case study with a level of nesting [23] which categorised Saraya district into urban zones and rural zones. These approaches are relevant to research that focuses on understanding (how?) and explaining (why?) complex human behaviours and phenomena occur [23]. The case studies were undertaken in one town (Saraya) and two villages (Sekoto Dantila and Sabadola) in Saraya district and in multiple Koranic schools in the urban district of Diourbel.

## Sampling strategy

Given that caregivers' perceptions were previously shown to be related to gender, age, prior experience and place of domicile [15, 17], we purposely selected respondents with different backgrounds. This sampling strategy led to the diversity of gender, age, prior experience and place of domicile, whether in a rural or an urban setting. Purposive sampling was directed towards achieving maximum variation in age and gender, using a snowball approach: 'a non-probabilistic form of sampling in which persons initially chosen for the sample are used as informants to locate other persons having necessary characteristics making them eligible for the sample [24]. A purposive sampling method was used to recruit 50 participants across the three villages in Saraya district. These included 30 mothers, 10 fathers, 10 grandmothers who were primary caregivers of pre-school and school aged children. Similarly, we randomly selected 50 participants who were Koranic teachers and leaders of the Koranic schools (ouztaz) and were also primary caregivers for the Koranic students (talibés) in these schools (dahras). Sample size was estimated to be sufficient based on the principle of theoretical saturation [24] and based on our previous experience with this methodology, we expected to reach saturation with approximately 50 interviews per group.

## Data collection methods

We used several methods to collect qualitative data from the study participants; these included individual and group interviews and participant observations. The final number of individual and group interviews was determined based on the saturation of the data that we obtained during the collection phase. Face-to-face interviews were facilitated by trained research assistants, using a purposed-designed interview guide (S1 Table). The interview guide explored the risk perception of factors that increased vulnerability to developing a combined infection of malaria and helminths in children. The study participants were asked about the sleeping habits of their children and whether they used a treated bed net, the structure of their house (whether the walls of the house were made of mud and poles, cement blocks, corrugated iron, concrete, mud bricks, burnt bricks, wood, and whether the roof was made of corrugated iron or grass), and toilet facilities. We visited the houses and inspected the surroundings and toilets. We explored a measure of the sanitation level of the households by assessing the toilet facilities and handwashing practices through direct observations of the study participants. The participants were asked to show where they and other members of the households most often washed their hands at any time, i.e. before a meal, before cooking or feeding a child, and after using a toilet. When no handwashing place was shown, the respondents were probed for reasons why they did not have the facility or practised handwashing. We observed whether water and soap were available at a designated handwashing location within the households. The households were also observed for the presence of soaps, or other cleansing agents within an arm's reach of the place for handwashing. The respondents were not asked to fetch soap, as this did not reflect the soap's accessibility.

In-depth interviews were also held with the primary caregivers of the pre-school and school age children in the households and Koranic schools to gain an in-depth understanding of their perceptions on risk of their children/wards developing mixed infections of malaria and helminthiasis. The respondents were asked about recent illnesses with malaria and helminths, and what they thought might make the children developed these diseases. They were asked about hypothetical illnesses to understand how health-seeking behaviour may change by illness type. The interviews were conducted in the local languages preferred by the participants and these were recorded using digital dictaphones.

## Data processing and analysis

The transcriptions of the interviews were carried out by trained research assistants who were native speakers of the local languages in Senegal. We verified the meaning and interpretations of the transcribed texts and confirmed consistency with the original texts in the local languages. The entire coding phase followed the inductive coding methodology. We organised the data around evocative themes with regard to the comments and views expressed by the study participants. We used thematic analysis through an objective and systematic analysis of the contents of oral discourse. We extracted the different units of analysis of the discourse and performed horizontal analysis of the theoretical framework in relation to the study objectives. We created themes that emerged from the analysis of the codified data, and validated the meaning by triangulation of the sources and methods of gathering the data. We analysed the data using NVivo 12 software and presented the results along the emerging themes.

## Study trustworthiness

In line with trustworthiness criteria created by Lincoln and Guba [25], we ensured credibility of the study by prolonged engagements, persistent observations, data collection triangulations, and researcher triangulations. Through an iterative process of listening, discussing, and re-listening, the research team identified and consensually validated emerging themes and appended segments of dialogues supporting the proposed themes. We stopped interviews when saturation was reached (i.e. when no new themes were identified). The team systematically reviewed the themes and sorted them into content domains. We used an analytic matrix to identify patterns and connections amongst the domains. Two of us not involved in the qualitative coding process (MA and IAM) audited the analytic matrix, choice of quotes, and thematic analysis.

The team members checked the descriptions of the key phenomena and themes which emerged from the data analysis with the study participants and requested them to verify the accuracy or consistency with their perspectives. These activities supported the validity and transferability of our study findings.

We kept robust records of the raw data, field notes, transcripts, and reflexive journals from this study and this helped us to seamlessly systemize, relate, and cross reference data. We maintained an audit trail that documented evidence of the decisions and choices that we made regarding the theoretical and methodological issues throughout the study period. These steps supported the dependability of our study [26].

To achieve confirmability [26], we highlighted the rationales for theoretical, methodological, and analytical choices throughout the entire study and this supported that the interpretations and findings were derived from the data collected from the study participants.

Reflexivity [26] was maintained by the research team through the analysis and writing by recording, discussing and challenging established assumptions. In addition, MOA, NMS and AD kept reflexive diaries. The first author observed the interviews and discussion groups. He

was not known to the participants of this research prior to undertaking the study as a medical doctor who had first-hand personal experience in the subject of discussion. Whilst it was useful to 'know' (from his own background) what the participants were talking about medically (and in terms of detecting items of significance), as a researcher he made conscious efforts not to accept potentially common assumptions at face value.

## Results

This study took place in June and December 2021 in Saraya and Diourbel districts, respectively. A total of 50 study participants was recruited in each district. The socio-demographic characteristics of the study participants are summarised in Table 1 below.

### Findings from the direct observations of the households

The majority of the 'dahras' visited had their bungalow houses built with blocks and roofed with corrugated zinc while slab roofs, cement blocks and iron doors were used for the two-storey buildings (Fig 2). In other 'dahras', the rooms which served as a place of study for the children were either made of straw, wood or zinc, with zinc roofs. In almost all the 'dahras', there was a large hall which served as a place of study for the students, without a fence but with a zinc roof and cement or tiled floor and sometimes the floor was only sand. None of the 'dahras' had mosquito netting fitted on the windows or doors. These observations were corroborated by comments from a senior student, a leader of the Koranic school and a wife of a school leader:

> "Yes, the whole building is solid, it's a new building and the whole floor has been tiled. The doors and windows are iron. There are no mosquito nets on the doors or on the windows. » Supervising Talibé

> "All the other rooms are made of zinc, the walls, the roof, the doors and windows. They are six in number (6), this is where the Talibés sleep. The floor is cement except for one of the bedrooms, and that's because the work hasn't finished yet. » Head of Dahra

**Table 1. Socio-demographic characteristics of study participants, Diourbel and Saraya, 2021.**

|  | Diourbel (n = 50) % | Saraya (n = 50) % |
| --- | --- | --- |
| **Age group** |  |  |
| 18–49 years | 35 (70) | 32 (64) |
| ≥50 years | 15 (30) | 18 (36) |
| **Gender** |  |  |
| Male | 41 (82) | 10 (20) |
| Female | 9 (18) | 40 (80) |
| **Relationship to children** |  |  |
| Mothers | - | 30 (60) |
| Fathers | - | 10 (20) |
| Grandmothers | - | 10 (20) |
| Koranic teachers | 30 (60) | - |
| Leaders of Koranic schools | 11 (22) | - |
| Wives of leaders of Koranic schools | 9 (18) | - |

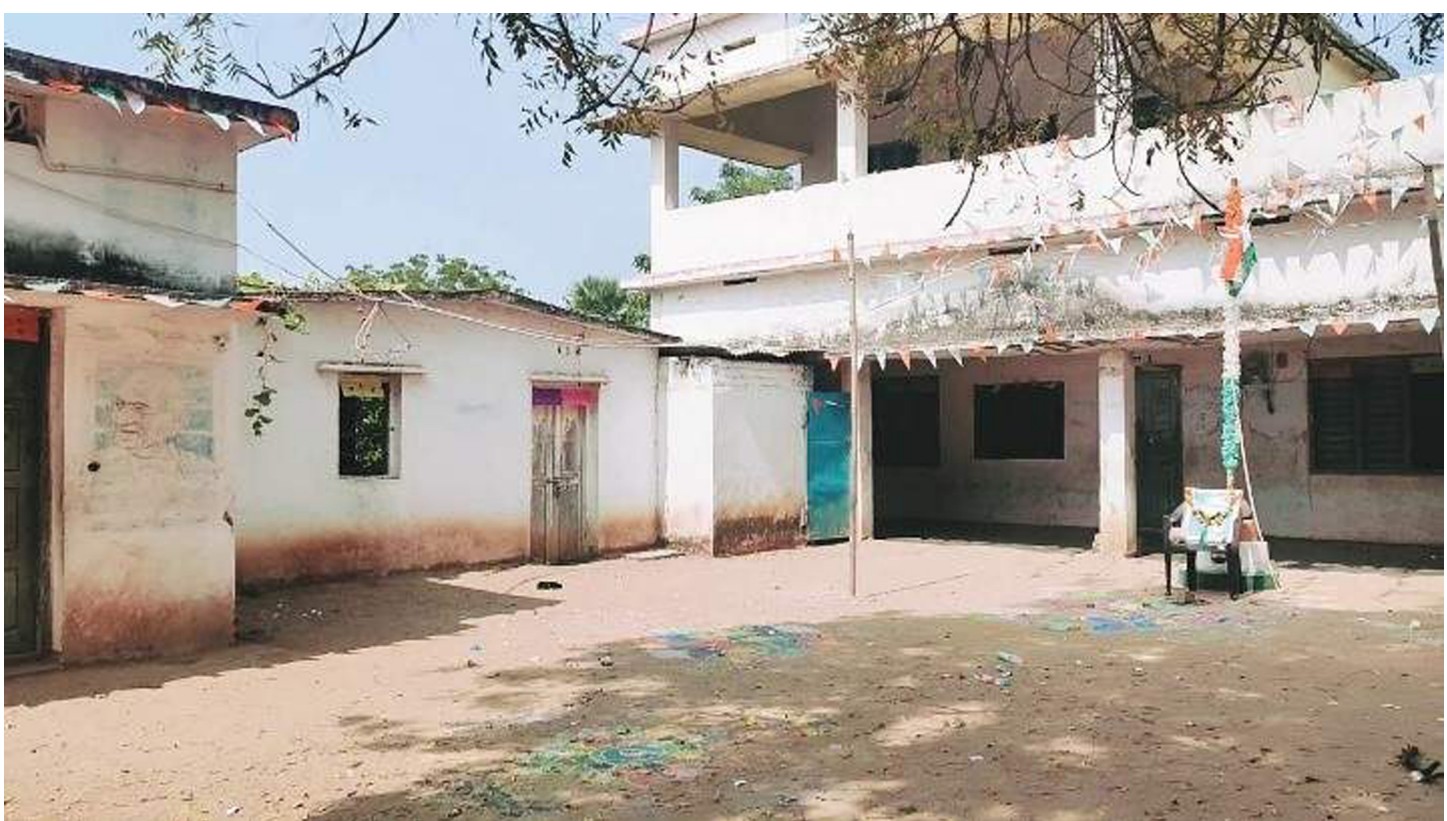

**Fig 2. Picture of a household in Diourbel site.**

*"It's a house in a barrack, a bedroom, a kitchen and a veranda, the wall is wooden and the floor is cement, a large hall where the Talibés (sand) study and sleep. The entire roof is in Zinc, the windows and doors are also in Zinc.* » Wife of a Dahra official.

Similarly, in most dahras, the children (talibés) lived in adjoining rooms, some of which had no doors. Also, we observed multiple holes in the roofs, which implied possible leakage of water through the roofs during rainy seasons. A senior student confirmed that they often put containers in each corner of the rooms to collect the rainwater and to prevent flooding the rooms or getting their clothes wet.

*"Yes, in the room where the children sleep, the water enters there during rain. Either they sleep with that, or they put pots to prevent it from touching them.* » Supervising Talibé

### Bedding organisation

In some 'dahras', talibés slept according to age groups. The younger children slept on the mats laid on the floors while the older ones slept on mattresses. Occasionally, the older ones slept under a mosquito net, installed by one of them. Most talibés preferred to sleep in the courtyards of the 'dahras' because of intense heat during hot seasons while others slept in the open spaces in front of the houses, for the same reason.

*"We have mats at our disposal which constitute beds for the learners. So these mats are given to the learners, according to their age groups, and the rooms, which are five in number. Those who are between 7 and 12 years old share the same room and those who are between 10 and 15 years old also share the same room, and finally, those who are between 15 and 20 years old are in the same room.* » Head of Dahra.

The household structure and bedding arrangements in the 'dahras' differed significantly from those found in Saraya villages. Majority of the houses in Saraya villages were made of mud walls and thatched roofs (Fig 3). Few had corrugated zinc roofs while none had mosquito netting on the windows or doors. Nevertheless, almost all caregivers in Saraya confirmed that they and their children slept under a mosquito net during the night preceding the interviews.

## Hand washing practices

All 'dahra' leaders and Koranic teachers reminded us of the sacred dimension of hygiene in the Islamic religion. The ablutions performed by muslims by washing different parts of the

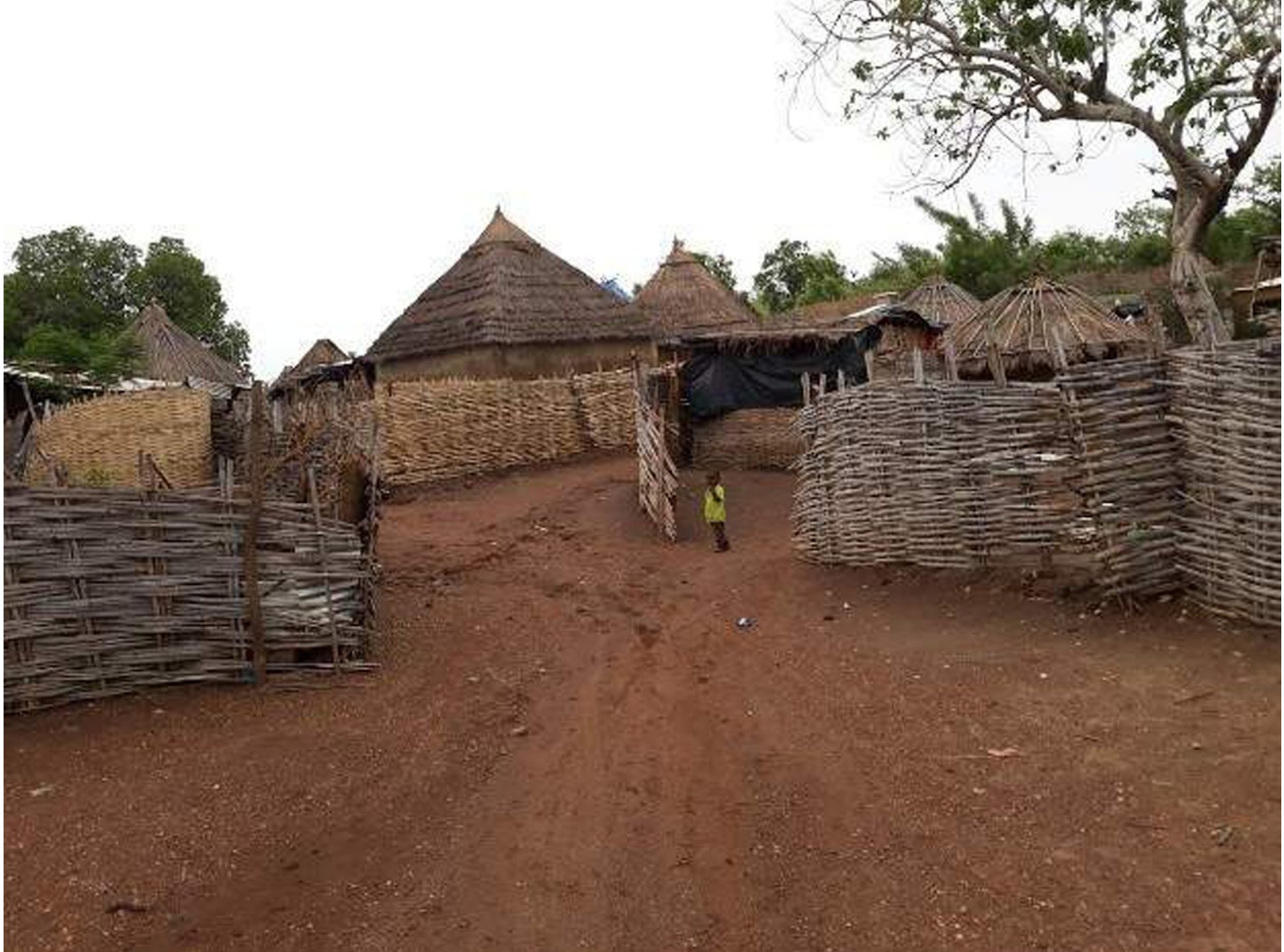

**Fig 3. Picture of a household in Saraya site.**

body with water before observing the mandatory five daily spiritual prayers, were cited as concrete examples of the importance given to hygiene and hand washing in Islam. They also affirmed that the concept of general body cleanliness and hand washing occupied an important place during daily discussion/awareness sessions organised for the students in the 'dahras'.

> *"I will answer you by Islamic jurisprudence because the prophet of Islam said that each person must wash himself every morning before doing anything because he does not know where his hands spent the night. So even before washing the body or doing ablutions, you have to wash your hands, all the more so when you decide to cook. Cleanliness is all-encompassing in Islam, everything that must enter the body and the belly must be clean"* Head of Dahra

The advent of Covid-19 outbreak facilitated the training of many 'dahra' managers on good hygienic practices. In addition to the training workshops, the district health authorities also provided the managers with products and accessories to perform hand hygiene. We also observed many pictorial flyers written in local languages, demonstrating the steps in handwashing that were displayed throughout different sections of 'dahras'. Although knowledge of the importance of good handwashing practices was well ingrained among the caregivers, including the health risks that could arise from poor hygiene, most of the interviewees acknowledged the difficulties they faced in enforcing strict compliance with these good practices on the children. Limited financial capacity and human resources were also cited as barriers to enforcing compliance with handwashing among the talibés. For example, when the talibes left the toilets, some talibés washed their hands routinely with water but rarely with soap, mainly because soap was not always available. In the absence of soap or detergent, a few talibés used sand to clean their hands after visiting the toilets. To the question "Is there "madar" (detergent) placed in front of the toilets? The response from many of the teachers was *"No, there isn't.* With what do they wash then, his answer was: *"Usually with water and sand"*.

> *"We taught them that (repeated many times), but there is no one who is there all the time to remind them, if we see them leaving the toilet without washing, we can call them to order, but there is no one who reminds them of these incessantly. »* Wife of a dahra teacher.

The descriptions obtained in Diourbel about handwashing were similar to the responses provided by the caregivers in Saraya villages, where majority considered hand washing useful to prevent a lot of diseases. Like their counterparts in Diourbel, Saraya participants also linked the importance of handwashing and body cleanliness to Islamic injunctions. The participants also highlighted how the public health messages shared as part of preventive measures for Covid-19 pandemic, reinforced their hand washing practices.

> *"Handwashing helps prevent disease. Here, almost all of us are used to washing our hands, adults and children alike, before eating everyone washes their hands on a pot with soap"*
>
> Mother, 29 years old, Sabadola

> *". . .it's because you won't be comfortable in your activities when you haven't washed your hands. It's not as pretty to see stepping out of the toilet and not washing your hands"*
>
> Mother, 28 years old, Saraya

As observed among Diourbel students, most participants in Saraya villages washed their hands with water or with a natural product such as ashes or sands. According to them, ashes can replace soap.

*"As soon as you leave the toilet, you have to wash your hands, before eating, wash your hands also if there is no soap. If this is not the case, use the ashes"*

Father, 32 years old, Sékoto Dantila

*"When we have money we can buy soap and when we don't have money we only use water like that to wash ourselves and even when leaving the toilet and this constitutes a risk of catching diseases »*

Mother, 50, Sekoto Dantila

## Toilet hygiene and toilet use practices

Most of the 'dahras' visited had modern toilets built with Turkish chairs, cement walls, a zinc roof and a septic tank as stool disposal system. Some 'dahras' had several toilets, shower toilets and stool toilets (Fig 4). In a few 'dahras', separate toilets were built for the Koranic teachers and the talibés. Traditional toilets built by digging pits with a seat made from cement, a straw fence and no evacuation system were also found in some 'dahras' located in the remote/bushy areas of Diourbel. These were similar to toilet facilities found in all households visited in Saraya villages. When these pit latrines were filled with stools, they were covered with sands and other pits were dug to serve as new toilets (Fig 5). In some households in Saraya, there were no toilets. In these cases, members of the households used neighbours' toilets or practised open defaecation.

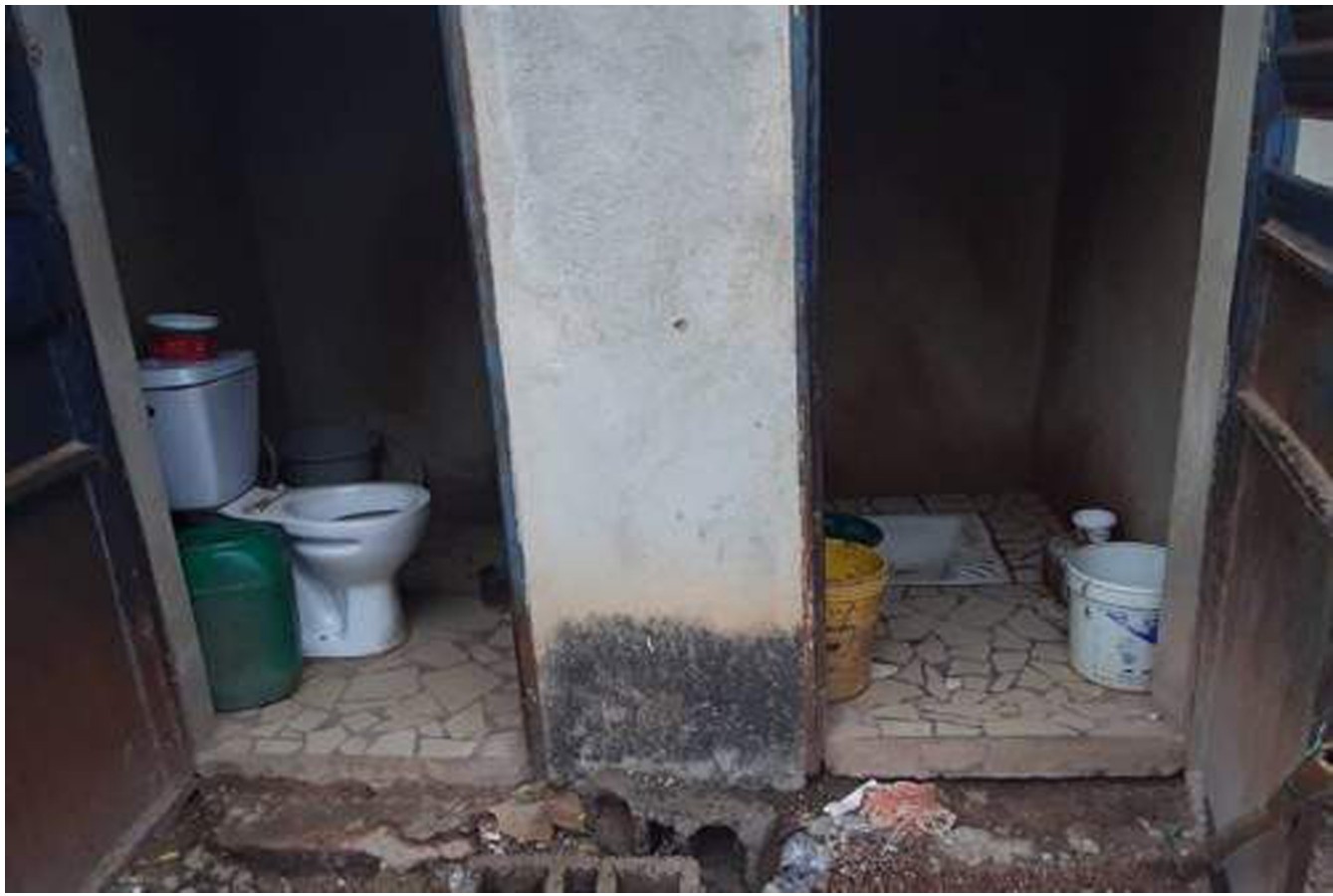

**Fig 4. Picture of a toilet facility in Diourbel site.**

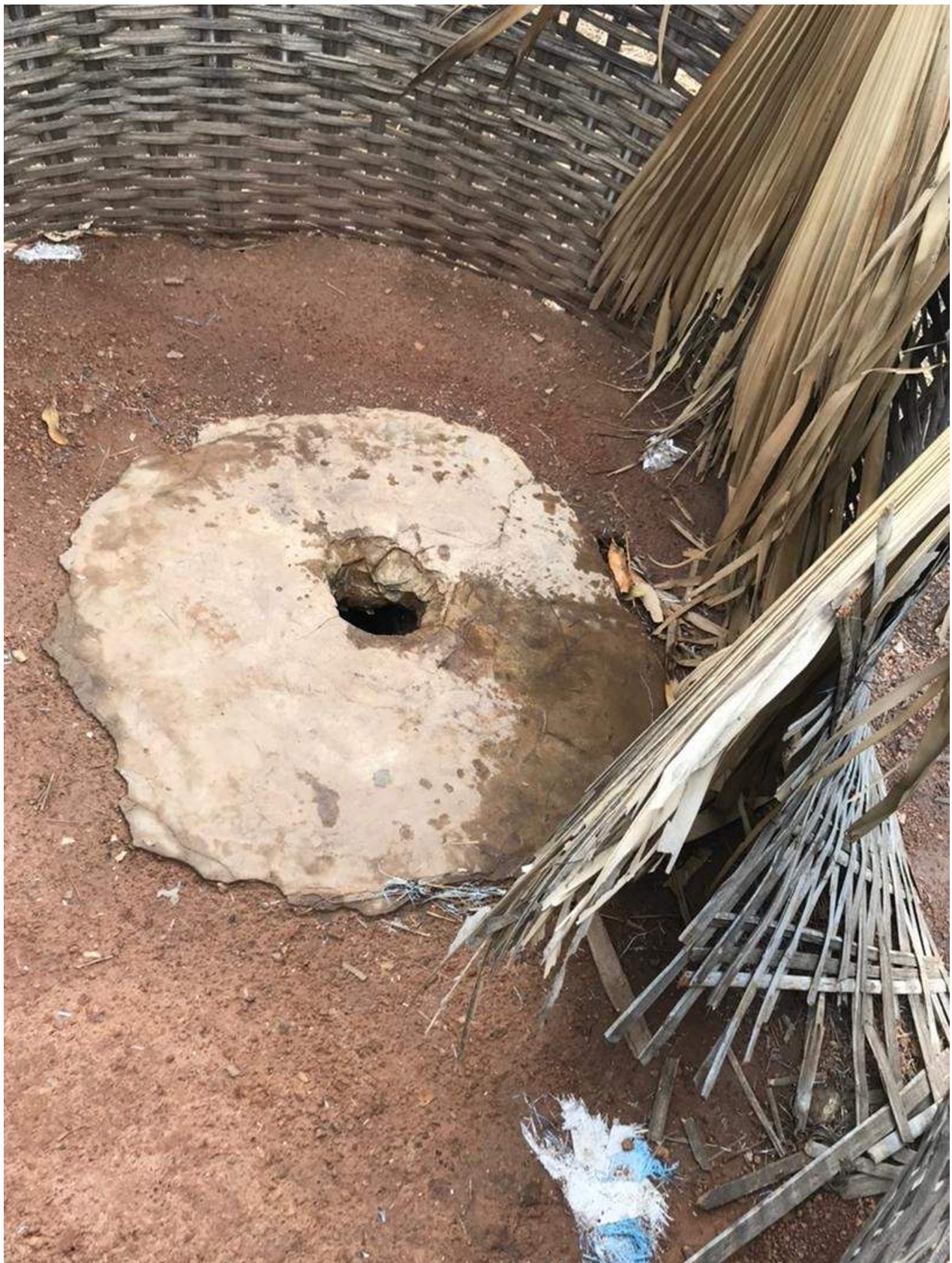

**Fig 5. Picture of a toilet facility in Saraya site.**

*"Here we have a traditional toilet fenced in with solid straw with bricks around it to act as a rampart. The water from the toilets pours behind the concessions or over holes and when it's full we close and dig another one, that's how we do it here"*

Mother, age unknown, Sekoto Dantila

*"We don't even have a toilet here, we go to the bush to do open defecation. »*

Grandmother, 60 years old. Sabadola

Most children aged under five years in Saraya villages used potties to pass stools, following which the mothers or caregivers dumped the stools in the home-made toilets or nearby bushes. The mothers cleaned these children with their bare hands, which were later washed with water alone or water and soap, depending on the availability of the latter.

*"Children who are under 5 years old use the pots . . .. As soon as the child wants to relieve himself, we give him a potty, if he finishes we clean it and the potty too, to throw the stool in the toilet".*

Grandmother, 59 years old, Saraya

## Practices related to swimming and barefoot walking

All participants interviewed in Diourbel and Saraya expressed concerns and difficulty in getting children to wear shoes. Owing to the itinerant nature of the Koranic students in Diourbel, the shoes get lost each time the Koranic leaders procured new ones for the students. In Saraya villages, lack of money to buy new shoes for children was cited as the major reason for children walking barefooted.

*"(Laughs) it is very difficult, if not impossible, to get children to wear the shoes. Even, yesterday there was a man who asked if it is necessary to buy shoes for the children since they don't wear them, and I told him to buy them and then we'll keep them for them. They wear when going out but do not fit with the shoes.* Head of Dahra

*"They very often wear plastic shoes, often it is the over-15s who wear them the most and small children are often used to being barefoot"* Head of Dahra

## Existence of swimming places

Our interviews showed that very few rivers and streams were available in Diourbel where children could bathe or swim. These rivers were very far from the 'dahras' and the students were strictly forbidden from going to the rivers for bathing or swimming, because of two episodes of drowning that occurred there. This situation contrasts sharply with Saraya villages, which had many rivers and streams where adults and children bathed, swam and washed their dirty clothes.

## Most frequent diseases

Malaria was cited by the majority of respondents in Diourbel and Saraya districts as the most common illness among the Koranic students and children. The students were also reported to complain a lot of stomach aches. Mosquito bites were mentioned by most respondents in both sites as the main cause of the recurrence and persistence of malaria, especially given that most Koranic students did not sleep under mosquito nets.

*"Malaria is the most recurrent. Apart from this pathology, children often suffer from stomach aches, and finally from colds, because you see children who cough, coughs from colds, it happens to them. »* Head of Dahra

*"Malaria is more common here and also stomach aches. The stomach aches are due to the bad food they consume once they go out. As they are talibés, they eat all the food they receive on the street. And these foods may not be good for them. »* Head of Dahra

*"It's the mosquitoes I think. But there is also the fact of not sleeping under a mosquito net, because there is no standing water here, so you can only think of not sleeping under mosquito nets as the origin. Recently I saw multiple cases of malaria in one of the houses, and they told me they didn't have mosquito nets, I think that's why. I spoke to MB about it and he told me that he will find a solution. »* Mother, unknown age, Saraya

## Management of malaria cases

Most of the Koranic masters and some of their wives were trained in the diagnosis of malaria using a rapid diagnostic test (RDT) and case management of malaria by the health authorities in Diourbel. The Koranic leaders were also provided with RDTs and Artemisinin-based combined therapy (ACT) for malaria treatment. These was no similar arrangement among the caregivers in Saraya villages, most of whom demonstrated good health-seeking behaviour and attributed the decline in malaria among the children to the yearly mass drug administration with seasonal malaria chemoprevention (SMC).

*"We have ACTs and malaria tests, so if we do the test and it comes back positive, we prescribe them the ACTs, respecting the dose, and the paracetamols. We order paracetamols from the main pharmacy. Now, if the test is negative, we give them paracetamol, because we will consider the signs like the flu. "* Daughter of a Koranic master.

*"Yes there were medicines that are given every month for three days. And after that, we saw that the malaria rate had dropped a lot, so I think that yes, certain drugs can help with malaria. But for the worms I don't really know maybe the marabout can answer you on that. »* Mother, 37 years, Sekoto Dantilla

## Knowledge and perceptions about worms

Unlike malaria which many dahra leaders could detect and manage at household level, the knowledge and perception about symptoms of worms and treatments were minimal. Only stomach ache was mentioned as the symptom of worm infestations by the Koranic teachers and they confirmed that this was rarely a reason for seeking treatments for the students in health facilities. In addition, some mentioned the students' poor diet and lack of hygiene as potential sources of worms, given that the children begged and received different kinds of food from the public. Amongst caregivers in Saraya, worm infestations were said to be caused by eating sugary food which could lead to having stomach ache and passing worms in stools.

*"We also think that children can have worms but it's not a lot, we have never had complaints about worms coming from children, there is no one who has come and that we consult him and that we realize that he has worms. For the worms we are the ones who assume that what the children eat is too sweet so they can have worms. »* Head of Dahra

*"They sometimes vomit or suffer from stomach aches. I remember my child was vomiting a lot, we went to the hospital, after the tests they did not see anything, they assumed it was worms and they prescribed medicine. »* Manager's wife

*"You see kids eating sugar cubes like that all the time, it can give them worms. In fact, not long ago, we took a child to the hospital and we were told that it was because of the sugar; his face was a little swollen. »* Grandmother, 58 years, Saraya

## Worm treatment

In Diourbel and Saraya, medications for the treatment of worms were given at a health post or during the mass deworming campaigns. Some reported that the drugs were not effective and caused side effects such as diarrhoea in children, without eliminating the worms. This led some to prefer to use home remedies.

*"Well, I don't know too much, but I know that the drugs we are given are to be given to the children every fortnight, that's all I know. For the most part, they recover but I admit that if he does not stop consuming certain foods, they will come back. »* Head of Dahra

*"The worms, we know it's complicated to treat, we know that we only have sedatives because at some point they come back (the worms). We know that killing them until they come completely out of the belly will be complicated. And these are things that also happen here. »* Head of Dahra

*"The worm medication works for some and not for others and for example last year we were given medication for this and it caused diarrhoea in some children. When he had diarrhoea, it was found that no worms came out. »* Mother, 29 years, Saraya

## Risk perception for malaria-worm co-infection

Whilst caregivers in Diourbel and Saraya demonstrated fair knowledge about the risk factors for malaria and worms as a separate disease entity, the possibility of a combined infection involving malaria and worms was considered to be a rare occurrence. The respondents in both sites emphasised that the symptoms and signs of malaria-worm co-infection were complex and difficult to recognise among the children in Saraya and Koranic students in Diourbel.

*"I never had the knowledge that a child can have malaria and worms together at the same time. We don't know if the child has worms, but if he has malaria, we know"* Supervising Talibé

*"You know that worms are inside the body and what causes it is excessive consumption of sugar like ice cream and sugar candies; children eat too much of it and it's so they have worms all the time. Malaria attacks anyone so it is also quite normal for a child to have malaria and worms at the same time.* Supervising Talibé

*"Worms are caused by the excessive consumption of sugar in children if it is added by mosquito bites they can have both diseases at the same time. »* Mother, 32 years, Saraya.

Some caregivers were of the opinion that children could suffer from malaria and worms at the same time, especially if children consumed too much sugary food and were later bitten by mosquitoes. There was also a consensus of opinion by the caregivers in Diourbel and Saraya

that the only way to know if a child has worms was to see the presence of worms in the child's stools.

> "Yes a child can have worms and malaria. When that happens, we have to go to the doctors, it is he who has the capacity to treat them to pass the worms in their stools. Having these two can be caused by excess sugar intake and the bite of a mosquito. » Supervising Talibé

> "I have never seen a child have malaria and worms, but I think a child can have both diseases at the same time. Worms are caused by the excessive consumption of sugar in children if it is added by mosquito bites they can have both diseases at the same time. » Mother, 28 years, Sabadola.

## Discussion

We assessed caregivers' perception about the risk for malaria and helminthiasis as a separate disease entity and malaria-helminth co-infection among pre-school and school-aged children living in the urban and rural communities that are endemic for both diseases. Our findings showed a very high level of awareness about the risk for malaria but almost all the study participants demonstrated poor knowledge of the risk and causative factors for worms in children. More importantly, most caregivers in Diourbel and Saraya did not perceive the possibility of the co-existence of infections involving both malaria and worms in their children. The relatively better perception of the risk and factors associated with malaria and its consequences demonstrated by a majority of the study participants is similar to findings reported in previous studies [13, 14]. Also, greater investments on community messaging associated with effective implementation of malaria control programmes in Senegal is likely to have contributed to the high perception and increased awareness about the risk and health-seeking behaviour for malaria, demonstrated by the study participants.

The training and support provided by the Senegal health authorities to the Koranic school teachers to diagnose and treat malaria at household levels were impressive, demonstrating the commitment of Senegal to achieving malaria elimination. An optimal, if not the same level of awareness about the risk of worm infestations would have been demonstrated by the participants, if similar support was given to NTD control.

Our study communities were characterised by socio-economic vulnerability and social stratifications on gender norms in relation to hygiene and sanitation. The management of hygiene of the households and that of the children rested largely on the shoulders of women and older students at the Koranic schools. In Saraya villages and in similar settings in African communities, women were responsible for collecting stools, cleaning up younger children and disposing their wastes while men were mainly responsible for providing financial support to procure materials such as soap to ensure hygiene within the household. Given the setting in 'dahras', older children who were students at the Koranic schools, were responsible for their own toilet management which was confirmed by their caregivers as not being hygienically sound. Despite being aware of these unhygienic practices, we expected the leaders and teachers at the 'dahras' to perceive this as a major risk for worm infestations and similar diseases which manifested as abdominal pain in the students. Situating the apparently low perception of the risk of worm infection demonstrated by the caregivers within the Health Belief Model showed that the perceived susceptibility of the children to worms was shaped by the caregivers' perception that worms were less harmful as a cause of serious illness in children than malaria. This perception may also downplay the need to take appropriate cues that may culminate in acceptable health-seeking behaviours [14]. Also, the caregivers' widely held opinions that excessive

consumption of sugary food was responsible for worm infestations in the children reinforces the relevance of the Health Belief Model in shaping the perception of risk and health-seeking behaviours.

Given the urban and rural locations of our two study sites, the household structures in Diourbel and Saraya were different not only in term of the materials used to build the walls and roofs, the bedding and sleeping arrangements, toilet types and toilet practices were also significantly different. We opined that if the caregivers had adequate knowledge and/or awareness about malaria-helminth co-infection, they could have probably perceived these factors as risks for the co-infection and mentioned them during the interviews. In the same vein, considerable risk posed by the practice of open defecation, dumping of stools in the bushes and use of traditional toilets without hygienic evacuation system [27, 28], were not highlighted or perceived by the caregivers as risks during the interviews.

Nevertheless, handwashing practices were well perceived by the caregivers across the two sites. The good knowledge of handwashing practices demonstrated by the caregivers was most likely influenced by the doctrines of Islamic religion [29, 30] which almost all the caregivers in Diourbel and Saraya professed. Whilst all study participants recognised the importance of handwashing with water and soap, non-availability of soap made the majority of them resort to using ashes and sands as alternatives to soap. Although the use of ashes and sand may sound practical because it was readily available at no cost, the eggs of the helminths are known to reside within sands and could promote the transmission of worms to the children, thereby increasing their risk and vulnerability to soil-transmitted helminths [28, 31].

Almost all study participants demonstrated poor perception about the risk for malaria-helminth co-infection, as they attributed the risk of the co-infection to a combination of excessive consumption of sugary food and mosquito bites. Studies have documented similar misconceptions about the risk and causes of common childhood diseases in Africa and how this had negatively impacted on health-seeking behaviour [32, 33]. Given that caregivers who perceived that their children could be susceptible to a disease were more likely to seek treatment compared to caregivers who had low perception about susceptibility to the disease [13, 34], there is a need to enhance the caregivers' knowledge on malaria-helminth co-infection and its associated complications in children.

Our study had a few limitations. Our findings were largely based on self-reports provided during the interviews by the caregivers. The respondents might wish to impress the interviewers, but the inclusion of participant observations reduced this bias. Also, the study populations in Diourbel and Saraya were not homogenously similar, hence, it was difficult to draw conclusions on some comparative findings obtained from the two diverse communities. Nevertheless, the selection of the two sites provided the opportunity to explore and understand the perspectives of the caregivers on risk factors for malaria-helminth co-infection in the paediatric populations living in different contexts within the same country. The use of multiple qualitative methods also enabled us to generate findings that reflect the diverse settings in a typical African community.

In conclusion, a majority of the caregivers in our study demonstrated a high perception of risks as well as acceptable health-seeking behaviour for malaria, but low risk perception and misconceptions about the causative factors of worms and malaria-helminth co-infection. The findings of this study underscore the need to promote awareness about the risk and complications of malaria-helminth co-infections in children. This step would assist in addressing the caregivers' misconceptions about the co-infection and may improve their uptake of the strategic interventions developed to achieve control and elimination of malaria and helminth co-infection.

## Supporting information

**S1 Table. Interview guide used to collect qualitative data from the caregivers.**
(DOCX)

## Acknowledgments

We appreciate Dr Babacar Gueye, Dr Doudou Sene, Dr Ndeye M'backé Kane, Dr Boubacar Diop and the entire management at the Senegal National Malaria Control Programme, NTD Control Programme, SMC Programme and the Ministry of Health and Social Action for supporting the implementation of this study.

## Author Contributions

**Conceptualization:** Muhammed O. Afolabi.

**Data curation:** Muhammed O. Afolabi, Ndèye Mareme Sougou, Aminata Diaw.

**Formal analysis:** Ndèye Mareme Sougou, Aminata Diaw.

**Funding acquisition:** Muhammed O. Afolabi, Brian Greenwood.

**Investigation:** Muhammed O. Afolabi, Ndèye Mareme Sougou, Jean Louis A. Ndiaye.

**Methodology:** Muhammed O. Afolabi, Ndèye Mareme Sougou.

**Project administration:** Muhammed O. Afolabi, Doudou Sow, Isaac A. Manga, Ibrahima Mbaye, Jean Louis A. Ndiaye.

**Resources:** Muhammed O. Afolabi.

**Supervision:** Muhammed O. Afolabi, Doudou Sow, Isaac A. Manga, Ibrahima Mbaye, Brian Greenwood, Jean Louis A. Ndiaye.

**Validation:** Muhammed O. Afolabi.

**Visualization:** Muhammed O. Afolabi, Brian Greenwood.

**Writing – original draft:** Muhammed O. Afolabi.

**Writing – review & editing:** Muhammed O. Afolabi, Ndèye Mareme Sougou, Doudou Sow, Brian Greenwood, Jean Louis A. Ndiaye.

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
