## [Decision Letter · Decision Letter 0]

11 May 2022

PGPH-D-22-00478

Caregivers' perception of risk for malaria, helminth infection and malaria-helminth co-infection among children living in urban and rural settings of Senegal: a qualitative study

Dear Dr. Afolabi

Thank you for submitting your manuscript to PLOS Global Public Health. After careful consideration, we feel that it has merit but does not fully meet PLOS Global Public Health’s publication criteria as it currently stands. Therefore, we invite you to submit a revised version of the manuscript that addresses the points raised during the review process.

We look forward to receiving your revised manuscript.

Kind regards,

Peter Bai James, PhD

Academic Editor

Journal Requirements:

1. Please provide additional details regarding participant consent. In the ethics statement in the Methods and online submission information, please ensure that you have specified whether: 1) whether the ethics committee approved the verbal/oral consent procedure, 2) why written consent could not be obtained, and 3) how verbal/oral consent was recorded. If your study included minors, please state whether you obtained consent from parents or guardians in these cases. If the need for consent was waived by the ethics committee, please include this information.

2. Please provide an Author Summary. This should appear in your manuscript between the Abstract (if applicable) and the Introduction, and should be 150–200 words long. The aim should be to make your findings accessible to a wide audience that includes both scientists and non-scientists. Sample summaries can be found on our website under Submission Guidelines: 

https://journals.plos.org/globalpublichealth/s/submission-guidelines#loc-parts-of-a-submission

3. Please amend your Data Availability Statement and indicate where the data may be found

Additional Editor Comments (if provided):

Here are some comments the authors should consider.

1.The authors need to clearly articulate the justification for conducting the study and how their findings contribute to the current scholarship

2. The authors need to discuss the theoretical framework underpinning their study. Was it an Ethnographic study or was it based on grounded theory or was it phenomenological? If no theoretical framework was used, it would be good for the authors to explain why.  Also, it would be good attached a copy of the interview guide

3. To ensure the validity of their findings, the authors need to discuss the trustworthiness of their study based on the following themes credibility, dependability, confirmability, transferability, and reflexivity.

<o:p>4. Please address reviewer 2 comments in the attached PDF document. </o:p>

Reviewers' comments:

Reviewer's Responses to Questions

**Comments to the Author**

1. Does this manuscript meet PLOS Global Public Health’s publication criteria? Is the manuscript technically sound, and do the data support the conclusions? The manuscript must describe methodologically and ethically rigorous research with conclusions that are appropriately drawn based on the data presented.

Reviewer #1: Partly

Reviewer #2: No

2. Has the statistical analysis been performed appropriately and rigorously?

Reviewer #1: No

Reviewer #2: No

3. Have the authors made all data underlying the findings in their manuscript fully available (please refer to the Data Availability Statement at the start of the manuscript PDF file)?

Reviewer #1: No

Reviewer #2: No

4. Is the manuscript presented in an intelligible fashion and written in standard English?

Reviewer #1: No

Reviewer #2: No

5. Review Comments to the Author

Reviewer #1: The author need to be concise on what was the aim of the study, what were the study findings. Currently, the manuscript has ambiguity who was interviewed (caregivers versus the respondents) Observational data is more subjective and it is skewed to the interpretation of the observer.

Was the author determining knowledge of Malaria, helminths and co-infection or the perception of care givers to the severity of these infections?

In the current format, the manuscript has no new information that can be published.

Reviewer #2: The manuscript on “Caregivers’ perception of risk for malaria, helminth infection and malaria-helminth co-infection among children living in urban and rural settings of Senegal: a qualitative study” by Afolabi et al. attempts to qualitatively address an issue of public health significance. The purported findings attempt to underscore the need to promote caregivers’ awareness about the existence and risk of malaria-helminth co-infection in children. This is very relevant in shaping control policies on malaria and helminth infections in the said settings and other endemic areas especially where poverty and poor sanitary conditions are the norms.

The authors in their paper have presented a narrative of their observations without any statistical analysis to back their findings. I acknowledge it is a qualitative study that notwithstanding the authors would have quantify the responses of the participants to carry out some statistical analysis to back their findings.

Secondly, participants in the study have not been adequately described for the readers to relate to the respondents and how the authors arrived at the purposive sampling of a total of100 participants (50 from each study area).

The intended audience would barely relate to the description of the housing structure and other unique set up described in the paper which is crucial to the understanding of the context if a graphical representation of these structures and map designating the various study sites are not included.

There are several lapses in the paper that makes repetitiveness of the study and the assertion of the findings practically impossible. I have made several comments on the paper for suggested improvement if the authors deem in necessary to publish in a journal of such a repute.

6. PLOS authors have the option to publish the peer review history of their article (what does this mean?). If published, this will include your full peer review and any attached files.

**Do you want your identity to be public for this peer review?** For information about this choice, including consent withdrawal, please see our Privacy Policy.

Reviewer #1: **Yes: **Nyamongo Onkoba, PhD

Reviewer #2: No

---

## [Decision Letter · Decision Letter 1]

18 Jul 2022

PGPH-D-22-00478R1

Caregivers' perception of risk for malaria, helminth infection and malaria-helminth co-infection among children living in urban and rural settings of Senegal: a qualitative study

Dear Dr. Afolabi,

Thank you for submitting your manuscript to PLOS Global Public Health. After careful consideration, we feel that it has merit but does not fully meet PLOS Global Public Health’s publication criteria as it currently stands. Therefore, we invite you to submit a revised version of the manuscript that addresses the points raised during the review process.

We look forward to receiving your revised manuscript.

Kind regards,

Peter Bai James, PhD

Academic Editor

Journal Requirements:

Additional Editor Comments (if provided):

In addition to responding to the reviewers' comments, do not forget to review the whole manuscript for typos and grammatical errors. Reviewer 2 comments are in the attached manuscript. Please confirm that all copyright issues have been resolved regarding the pictures you presented. Please provide a supplementary file that shows the current study adheres to the COREQ guideline for reporting qualitative research. In as much as local authors were involved in this study, it would be good to fill a copy of PLOS’ questionnaire on inclusivity in global research to explain how the local community was involved in this research. You can access the form via this link Best Practices in Research Reporting | PLOS Global Public Health

Reviewers' comments:

Reviewer's Responses to Questions

**Comments to the Author**

1. If the authors have adequately addressed your comments raised in a previous round of review and you feel that this manuscript is now acceptable for publication, you may indicate that here to bypass the “Comments to the Author” section, enter your conflict of interest statement in the “Confidential to Editor” section, and submit your "Accept" recommendation.

Reviewer #1: All comments have been addressed

Reviewer #2: All comments have been addressed

2. Does this manuscript meet PLOS Global Public Health’s publication criteria? Is the manuscript technically sound, and do the data support the conclusions? The manuscript must describe methodologically and ethically rigorous research with conclusions that are appropriately drawn based on the data presented.

Reviewer #1: Partly

Reviewer #2: Partly

3. Has the statistical analysis been performed appropriately and rigorously?

Reviewer #1: No

Reviewer #2: N/A

4. Have the authors made all data underlying the findings in their manuscript fully available (please refer to the Data Availability Statement at the start of the manuscript PDF file)?

Reviewer #1: No

Reviewer #2: Yes

5. Is the manuscript presented in an intelligible fashion and written in standard English?

Reviewer #1: No

Reviewer #2: No

6. Review Comments to the Author

Reviewer #1: Title: It should be trimmed down to " Caregivers' perception of risk of malaria and Soil transmitted helminths. It should not be pegged on either mono-infection or co-infection.

Abstract:

1. Avoid using the word "worms" specify whether it is STHs or tissue-dwelling helminths being referred to in the article.

2. Rephrase Ln 24 - 26; to bring out the congruency/co-endemicity of malaria parasites and helminths.

3. The aim focuses on co-infections, does it mean that mono-infections have been excluded from the study? Since they appear on the title.

4. What is the experimental design used? The inclusion criteria used in the study to recruit participants?

5. In LN 36 "a" before majority.

6. The study findings are ambiguous, How did the author measure perception? Refer to Ln 36 and 37 where the author indicates that the major study finding was that there was a high level of perception. How did they quantify/qualify this?

7. Is awareness different from knowledge? How was awareness determined in the study?

8. Ln 39 to 41 is vague.

9. The conclusion is not convincing. Let the author focus on sensitization and health promotion as key study recommendation.

Background Information.

1. The author fails to establish the niche and eventually occupying it in the study.

Overall, The narrative is anchored on the subjectivity of the author, there is no supporting evidence to make solid conclusions.

Reviewer #2: See comment on the manuscript attached for minor corrections.

7. PLOS authors have the option to publish the peer review history of their article (what does this mean?). If published, this will include your full peer review and any attached files.

**Do you want your identity to be public for this peer review?** For information about this choice, including consent withdrawal, please see our Privacy Policy.

Reviewer #1: **Yes: **Nyamongo Bw'Onkoba, PhD

Reviewer #2: **Yes: **Irene U.N. Sumbele

---

## [Decision Letter · Decision Letter 2]

25 Jul 2022

Caregivers' perception of risk for malaria, helminth infection and malaria-helminth co-infection among children living in urban and rural settings of Senegal: a qualitative study

PGPH-D-22-00478R2

Dear Afolabi,

We are pleased to inform you that your manuscript 'Caregivers' perception of risk for malaria, helminth infection and malaria-helminth co-infection among children living in urban and rural settings of Senegal: a qualitative study' has been provisionally accepted for publication in PLOS Global Public Health.

Best regards,

Peter Bai James, PhD

Academic Editor

Reviewer Comments (if any, and for reference):

Reviewer's Responses to Questions

**Comments to the Author**

1. If the authors have adequately addressed your comments raised in a previous round of review and you feel that this manuscript is now acceptable for publication, you may indicate that here to bypass the “Comments to the Author” section, enter your conflict of interest statement in the “Confidential to Editor” section, and submit your "Accept" recommendation.

Reviewer #1: All comments have been addressed

2. Does this manuscript meet PLOS Global Public Health’s publication criteria? Is the manuscript technically sound, and do the data support the conclusions? The manuscript must describe methodologically and ethically rigorous research with conclusions that are appropriately drawn based on the data presented.

Reviewer #1: Yes

3. Has the statistical analysis been performed appropriately and rigorously?

Reviewer #1: N/A

4. Have the authors made all data underlying the findings in their manuscript fully available (please refer to the Data Availability Statement at the start of the manuscript PDF file)?

Reviewer #1: Yes

5. Is the manuscript presented in an intelligible fashion and written in standard English?

Reviewer #1: Yes

6. Review Comments to the Author

Reviewer #1: (No Response)

7. PLOS authors have the option to publish the peer review history of their article (what does this mean?). If published, this will include your full peer review and any attached files.

**Do you want your identity to be public for this peer review?** For information about this choice, including consent withdrawal, please see our Privacy Policy.

Reviewer #1: **Yes: **Nyamongo Onkoba
